# The Effect of Biological Corrosion on the Hydration Processes of Synthetic Tricalcium Aluminate (C_3_A)

**DOI:** 10.3390/ma16062225

**Published:** 2023-03-10

**Authors:** Michał Pyzalski, Agnieszka Sujak, Karol Durczak, Paweł Murzyn, Tomasz Brylewski, Maciej Sitarz

**Affiliations:** 1Faculty of Materials Science and Ceramics, AGH University of Science and Technology, Al. Mickiewicza 30 Street, 30-059 Kraków, Poland; michal.pyzalski@agh.edu.pl (M.P.);; 2Department of Biosystems Engineering, Faculty of Environmental and Mechanical Engineering, Poznań University of Life Sciences, Wojska Polskiego 50 Street, 60-627 Poznań, Poland

**Keywords:** ordinary Portland cement, C_3_A, hydration, biocorrosion

## Abstract

This paper presents a study related to the biological degradation of a tricalcium aluminate (C_3_A) phase treated with reactive media from the agricultural industry. During one month of setting and hardening, synthetic C_3_A was subjected to corrosion in corn silage, pig slurry and chicken manure. The hardening process of the C_3_A phase in water was used as a reference sample. The phase composition and microstructure of the hydrating tricalcium aluminate slurries were characterised by X-ray diffraction (XRD), thermal analysis (DTA/TG/DTG/EGA), scanning microscopy (SEM, EDS) and infrared spectroscopy (FT-IR). In the samples studied, it was observed that the qualitative and quantitative phase composition of the synthetic tricalcium aluminate preparations changed depending on the corrosion exposure conditions. The main crystalline phases formed by the hydration of the examined samples in water as well as in corrosive media were the catoite (Ca_3_Al_2_(OH)_12_) and hydrocalumite (Ca_2_Al(OH)_7_·3H_2_O) phases. Detailed analysis showed the occurrence of secondary crystallisation in hydrating samples and the phases were mainly calcium carbonates (CaCO_3_) with different crystallite sizes. In the phase composition of the C_3_A pastes, varying amounts of aluminium hydroxides (Al(OH)_3_) were also present. The crystalline phases formed as a result of secondary crystallisation represented biological corrosion products, probably resulting from the reaction of hydrates with secondary products resulting from the metabolic processes of anaerobic bacterial respiration (from living matter) associated with the presence of bacteria in the reaction medium. The results obtained contribute towards the development of fast-acting and bio-corrosion-resistant special cements for use in bioenergetics.

## 1. Introduction

The degradation of cement concrete under progressive biological degradation is a common phenomenon in effluent or manure flow and storage systems such as pipelines, sewers and treatment facilities, which include digesters at biogas plants [1]. The progressive degradation of concrete sewer and wastewater pipes results in costly repairs and maintenance. The cost of restoring damaged sewer systems in Germany is estimated at approximately EUR 100 billion a year [2]. In Los Angeles, around 10% of sewer pipes are damaged by corrosion and the cost of restoration reaches USD 400 million. In the U.S., it is estimated that over the next 20 years, investment in the restoration and repair of existing facilities damaged by biological corrosion will reach USD 390 billion [2].

Raw sewage or manure contains a significant amount of sulphate ions, which are converted to hydrogen sulphide or other organic compounds produced under anaerobic digestion conditions [3]. In the sewage as well as in the reactors, part of the hydrogen sulphide reacts with atmospheric oxygen to form elemental sulphur, sulphites and thiosulphates, which can be deposited on concrete surfaces, resulting in their rapid uptake by sulphur-oxidising bacteria [4]. In addition, the characteristics of the chemical processes during the decomposition of organic fertilisers (manure, slurry) and the compounds formed during the methane fermentation reaction will adversely affect the durability of cement concretes [5,6,7].

During fermentation processes, the main components are methane, carbon dioxide and small amounts of nitrogen, hydrogen sulphide and hydrogen, as well as ammonia [8]. The course of methane fermentation is linked to a series of interacting stages. The subsequent stages produce organic chemical compounds, including products of the acidogenic (propionic acid), acetogenic (bicarbonates, acetates, alcohols) and methanogenic phases [9]. Sulphur compounds converted to sulphuric acid, as well as anaerobic methane fermentation processes, cause concrete degradation [9].

The natural effect of an increase in acid concentration associated with a reduction in the pH of the cement matrix environment in concrete is its reaction with the calcium hydroxide in the mortar/cement paste, leading to the formation of gypsum [10].

A decrease in the pH of the cement matrix environment will also affect the decomposition process of hydrated calcium silicates (C-S-H phase), with the effect that the solution will tend to equalise the concentration gradient of calcium ions, which will subsequently lead to the degradation process of the cement hydration products [11]. Other clinker phases, such as tricalcium aluminate and calcium alumino-ferrite, are involved in the calcium ion release processes and will release aluminium ions into the solution as a result of changes in the solution reaction. In addition, aluminium hydroxides and iron hydroxides will be formed. As a result of the corrosion reaction of concrete under sulphate ion conditions, secondary ettringite can be formed in the deeper parts of hardened concrete, where the pH remains relatively high. However, this process will result in expansion reactions in the concrete, resulting in micro-cracks opening up deeper parts of the cement concrete and the continuation of the progression of corrosion processes and, consequently, its total destruction [12,13].

Previous work carried out in the area of the biological corrosion of cement pastes held in aqueous animal waste solutions has shown that it is possible for thaumasite corrosion to occur, conditioned by an increase in the amount of the calcium carbonate phase formed as a result of the reaction processes of anaerobic bacteria respiration with calcium hydroxide [5,6]. In addition, studies have shown the possibility of counteracting the formation of corrosion products by using additives to seal the cement concrete matrix through the addition of calcium gamma orthosilicate, which is characterised by a significant enhancement in the specific surface area [7].

Tricalcium aluminate is commonly found in Portland clinker in amounts ranging from 3% to 15%; moreover, the course of its hydration process is similar to those that occur during the setting of aluminium cement (CAC) [10].

The above-mentioned phase and its hydration products in Portland cement (OCP) have not been studied for their resistance to biological corrosion conditions. Model studies of the above phase will allow a broader understanding of the mechanisms and progress of the chemical processes occurring at the interface between the C_3_A phase and the biological corrosion occurring during the exposure of this phase in aqueous solutions of animal and agricultural wastes.

The objective of this study was to investigate the hydration processes of tricalcium aluminate model phase samples deposited in aqueous solutions of biological wastes such as manure, pig slurry and corn silage. The results obtained were compared with controls such as non-treated samples or samples treated exclusively with water. It is expected that, in the future, the results obtained will allow the development of a special cement dedicated to construction projects related to the development of biofuels and to ecological development in agriculture [14].

## 2. Materials and Methods

### 2.1. Preparation of Components for Synthesis of Samples

Pure (99.9%) analytical reagents of calcium carbonate (CaCO_3_) (Avantor Performance Materials Poland S.A, Gliwice, Poland) and aluminium oxide (αAl_2_O_3_) were used to synthesise the C_3_A phase.

Preparation of the raw materials included preventive homogenisation and mixing of the raw material portions (1–3 kg, according to stoichiometric ratios used in a further part of the study) in a Deval drum (40 rpm) over a period of 36 h and drying in an oven (Czylok, Jastrzębie Zdrój, Poland) at 110 °C for a period of 24 h. Components prepared as above were sealed tightly in 10-litre plastic containers for further operations. Representative portions of each of the two ingredients were checked for their roasting losses by heat treatment at 1300 °C for 3 h. The roasting losses were 43.73 and 1.94% for CaCO_3_ and Al_2_O_3_, respectively.

The respective quantities of the individual ingredients (adjusted for roasting losses) were weighed out (Axis, AG500C) and the portions of raw material were placed in plastic containers (2L) with the addition of Ø = 7 mm balls made of hardened rubber to intensify the mixing process and prevent the secondary segregation of the components of the raw material and then homogenised over a period of 72 h in a Deval drum (40 rpm). The raw material was used for the temperature synthesis of tricalcium aluminate.

### 2.2. C_3_A Phase Synthesis

The molar mass of tricalcium aluminate is 270.18 kg/mol and consists of 62.3% calcium oxide and 37.7% alumina (Figure 1) [15].

Tricalcium aluminate decomposes at 1513 °C below its melting point through the elimination of calcium oxide with the simultaneous formation of the liquid phase in the system. In the experiment, the calcium oxide (CaO) content was assumed to be 62.5% (slightly above the stoichiometric content) to prevent the formation of mayenite (C_12_A_7_), which is thermodynamically the most stable phase in the CaO-Al_2_O_3_ system and de facto its formation cannot be fully eliminated.

Sintering of the pre-compressed samples took place at 1450 °C for 4 h. After removing the samples from the furnace, the resulting sinter was ground in an agate mill to a grain size corresponding to a 0.01 mm sieve mesh. The powder was then subjected to XRD analysis to confirm the presence of a cubic C_3_A phase.

### 2.3. Preparation of Samples for Testing

The samples were ground to a size at which the grains passed freely through a 0.01 mm sieve. This process was carried out in two stages: after the initial grinding of the preparations in a hand agate mill (Frisch, Germany), the water was removed by means of a rotary pump (Koszalin, Poland) under a vacuum [10^−3^ Tr] for a period of 1 h. This was followed by the second grinding of the samples in an agate mill and removal of unbound water for a further 24 h.

A 5 g sample was then compressed in a punch press PR-25R (Metimex, Pyskowice, Poland) with a matrix of a 15 mm diameter at a pressure of 1 bar into a cylindrical shape (6 mm height). Samples prepared in this way were placed in aqueous solutions of biological media. The temperature of the aqueous media solutions oscillated around 20 °C. Untreated anhydrous C_3_A (S) was prepared in order to study the effect of hydration in water. Over the 28-day experimental period, the anhydrous sample was kept in a desiccator.

Pig slurry (Tuszyn, Poland) (pH = 7.4) obtained from the industrial fattening of pigs, corn silage (Smolice, Poland) (pH = 6.2) and chicken manure (Fermy Drobiu Woźniak Sp. z o. o., Rawicz, Poland) (pH = 4.9) were used as active media for biocorrosion. The above biological materials in a quantity of 10 kg were diluted with 10 L of distilled water. The aqueous solutions thus prepared were subjected to maturation for a period of 14 days in a sealed plastic barrel.

The 5 g compressed samples were placed in sealed containers filled with water (sample W-control) or corrosive media such as aqueous solutions of pig slurry (G), corn silage (K) and chicken manure (P) and subjected to hydration. A sample subjected to tap water (pH = 7.3 ± 0.1) was used as a reference (control). The samples were kept in water or in the corrosive media at a constant temperature of 20 °C for 28 days. After this period, the samples were subjected to the removal of unbound water in a vacuum device [10^−3^ Tr] for 1 h and subjected to further tests.

### 2.4. SEM Images and EDS Spot Analysis

The morphology and chemical composition of C_3_A samples were examined on selected areas on the surfaces of the samples using the NOVA NANOSEM 200 (FEI Europe Company, Eindhoven, The Netherlands) scanning electron microscope coupled with an X-ray energy-dispersive spectroscope (EDAX).

### 2.5. XRD Measurements

The phase composition was examined using the X-ray diffraction method (XRD) carried out on powdered samples. At the end of the 28-day experimental period, the samples were ground once again according to the previously described methodology. Ground samples of approximately 1 g were pressed into a flat measuring holder and immediately tested to avoid the possibility of any prolonged contact with moisture.

The apparatus was equipped with a power supply stabilizing the operation of the PW 1140/00/60 X-ray tube and a modernised vertical PW 1050/50 goniometer (Philips Research, Eindhoven, The Netherlands). The goniometer was equipped with software for the full automatic computer control of the operation with the simultaneous digital recording of measurement data. The device was supplied with a vertical Philips X-ray tube with a Cu anode (Kα = 1.54178 Å); a Ni filter was used for the measurements. The XRD apparatus was equipped with a PW 2216/20 “fine focus” X-ray tube with a power of 1.0 kW (the lamp power used was 1.2 kW, which corresponded to a lamp voltage of 35 kV and a cathode filament current of 16 mA). A window of 0.4 × 8 mm (focal area of 3.2 mm^2^) at an incidence angle of 6° was used, providing a radiation beam with a width of 0.05 mm.

The obtained results of the quantitative mineral composition of the analysed samples were processed with specialised programs using the “Rietveld” algorithm [16]. One of these was the X’Pert High Score Plus software (Version 2.1 (2.1.0)) from “Philips” [17].

### 2.6. Thermal Analysis

Simultaneous Thermal Analysis (STA) was carried out on samples vacuum-dried and ground to a grain size below 0.010 mm. The tests were carried out using the DTA-TG thermal analysis method together with EGA analysis of the released gases. Measurements were carried out using a STA 449F3 Jupiter thermal analyser (Netzsch, Selb, Germany) coupled to a QMS 403C Aëolos quadrupole mass spectrometer (Netzsch). Measurements were carried out in alumina crucibles (Al_2_O_3_), on samples of approximately 75 mg, in a synthetic air atmosphere with a flow rate of 40 mL/min, at a heating rate of 15 °C/min, in the temperature range 30–1000 °C.

### 2.7. FTIR Measurements

The FTIR spectra of powdered samples were recorded with a Vertex 70v (Bruker, Billerica, MA, USA) spectrometer. A total of128 scans were collected for each measurement in the middle infrared (MIR) range of 4000–400 cm^−1^ with a resolution of 4 cm^−1^. For measurements, the test samples were mixed with KBr (1:400 mg) and then pressed into pellets under a pressure of 9 MPa.

## 3. Results

### 3.1. SEM Images with EDS Spot Analysis

Figure 2 shows SEM micrographs at a magnification of 350× of the transverse fracture of samples of tricalcium aluminate (S) exposed to the corrosive environments for 28 days at 25 °C.

Figure 3 shows a morphological image of the C_3_A sample (W) after exposure in water at a magnification of 5000×.

The hydration product of C_3_A in the form of a regular phase (point “2”—Figure 3) takes the form of spherical globules with sizes ranging from approximately 0.5 μm to approximately 4 μm. The second roentgenographically detected phase is very well-formed thin hexagonal plates (points “1” and “3”—Figure 3) with sizes ranging from approximately 8 μm to approximately 10 μm.

Exposure of tricalcium aluminate to a pig slurry environment leads to noticeable changes in its morphological structure (Figure 2c) compared to exposure to water. These differences can be seen most clearly in Figure 3 and Figure 4, which show high-magnification (5000×) SEM microphotographs and EDS analysis spectra of the chemical composition from selected areas of the above samples.

Sample G is dominated by two phases, namely the regular phase of C_3_AH_6_ and the hexagonal phase of C_4_AH_19_ (points “1” and “2”—Figure 4), which are morphologically similar to the hydration products of tricalcium aluminate in a water environment (Figure 3). In addition to these phases, calcium carbonate grains of the calcite variety were also recognised (point “3”—Figure 4), which are probably growing on the hexagonal hydration products of C_3_A, with sizes ranging from approximately 1.5 μm to approximately 4.5 μm.

Figure 5 shows SEM microphotographs of tricalcium aluminate held in corn silage (K), and examples of the EDS spectra spot analysis of the chemical composition from its selected areas.

Sample K shows large grains of regular catoite (point “1”—Figure 5) with sizes between 1.5 μm and 6.5 μm and aluminium hydroxide particles in the form of gibbsite. The presence of these phases was confirmed roentgenographically. A fine-grained microstructure is visible around the catoite phase, which is composed of calcium carbonate particles (point “3”—Figure 5). Occasionally, crystallites with a hexagonal structure are present, which usually form conglomerates of multiple hydration phases including calcium carbonate (point “2”—Figure 5).

After exposure to chicken manure, different morphological structures (Figure 2b) compared to the sample stored in water (Figure 2a) are observed. The SEM microphotograph shown in Figure 6 and the results of the EDS spot analyses of this sample indicate a number of crystallites in the form of thin filaments with a diameter of approximately 0.5 μm and a length of approximately 6 μm, as well as small-sized wavy separations of around 3 μm. In addition, spherical globular crystallites with diameters ranging from approximately 0.2 μm to approximately 1 μm can be found, attributed to a phase named catoite, and aluminium hydroxides as gibbsite of analogous mineral composition Al_2_O_3_·3H_2_O, as indicated by the results of EDS spot analyses (point “2”—Figure 6).

Considering the shape of the wavy precipitates and their chemical composition (point “1”—Figure 6), it can be reasonably stated that these are presumably pseudo-crystalline or amorphous forms of aluminium hydroxide. The thin fibres rich in aluminium and calcium may be derived from compounds containing phosphorus and sulphur. The origin of magnesium, phosphorus and sulphur can be explained by the fact that they are the main constituents of chicken manure. The significant amount of carbon, on the other hand, may indicate the presence of calcium carbonate particles in dense monolithic grain agglomerates, as shown by X-ray studies of this sample.

### 3.2. XRD Measurements

Samples of the anhydrous C_3_A phase (sample S—not treated), held in water (W—control, subjected to water) and subjected to hydration and biocorrosion in the presence of pig slurry (G), corn silage (K) and chicken manure (P), were examined to determine the qualitative and quantitative composition.

Figure 7 shows the XRD spectrum of a C_3_A anhydrous phase.

Two crystalline phases were detected in the qualitative composition of synthetic C_3_A (S). The dominant phase is cubic tricalcium aluminate (97.5%), accompanied by a small amount of C_12_A_7_ majenite (2.5%).

Qualitative studies of samples subjected to hydration and exposure in aqueous solutions of corrosive media are summarised in a collective diffractogram (Figure 8).

Differences in the diffractograms of the tested samples in a number of 2Θ ranges can be observed. Analysis of the 5–15 2θ region indicates changes in qualitative and quantitative composition depending on the storage conditions of the samples (Table 1). Differences in diffractograms in the 15–35 2θ region indicate the occurrence of new crystalline phases, which are formed as a result of biological corrosion processes in the tested samples. The sample hydrated in water (W) showed the presence of three crystalline phases, i.e., catoite—C_3_Al_2_(OH)_12_ (95%) and a hydrocalumite—Ca_2_Al_2_O_5_·8H_2_O (4%), accompanied by a small amount of unreacted synthetic tricalcium aluminate (approx. 1%).

Qualitative analysis of the sample subjected to pig slurry (G) showed that the catoite phase is present in the highest amount (44.5%). An additional high-intensity diffraction reflection at low 2θ angles indicates the presence of hydrated calcium aluminate—Ca_2_Al_2_O_5_·8H_2_O. A CaCO_3_ (3%) phase was also detected in the diffraction pattern of this sample. 

### 3.3. Thermal Analysis

Figure 9, Figure 10, Figure 11, Figure 12, Figure 13, Figure 14, Figure 15 and Figure 16 show the thermal curves of the DTA/TG/DTG signal as a function of temperature and, as complementary plots, the gas release curves of the ionic current for volatile phases with *m*/*z* 18 and 44 as a function of temperature.

The curve of anhydrous tricalcium aluminate showed no exo- and/or endothermic effects (Appendix A). These results were confirmed by TG and DTG curves, with no significant mass changes as a function of temperature in this sample.

Analysis of the gas release curves of synthetic C_3_A (Appendix A) shows the release of small amounts of water in the temperature range from 50 °C to 250 °C. This can represent the binding of water molecules from the ambient air due to the sample’s high hygroscopicity. No CO_2_ release was observed.

Primary analysis of samples indicates two groups of thermal curves. In group one, the similarity is shown by samples hydrated in water (W) and in pig slurry (G), while in group two, it is indicated by samples from aqueous solutions of chicken manure (P) and corn silage (K).

Analysis of the DTG and gas emission curve indicates that the thermal effects are mainly related to the release of water during sample heating (Figure 9 and Figure 10). Three types of effect can be distinguished: the first two have maxima occurring at around 90 °C and 190 °C, while the third effect appears as a linear increase in the release of water, with emission beginning at 230 °C and ending at approximately 590 °C. The results indicate at least two (or more) hydrated calcium aluminate compounds with varying chemical bond strengths. A very similar course of thermal curves is observed for C_3_A treated with pig slurry (G, Figure 11).

The DTG curve shows five endothermic peaks at temperatures of 86.2, 184.9, 329.8, 482.2 and 776.6, accompanied by weight losses of 1.22%, 3.22%, 18.28%, 3.05% and 4.05% (Figure 11), totalling 30.27%. A comparison of the results of the thermograms and the obtained data related to gas emissions indicates the presence of four effects associated with the release of gaseous components (Figure 12). The release of water as a result of heating the sample is associated with three effects, two of which occur in the temperature range from 100 °C to 200 °C. In the temperature range from approximately 200 °C to 350 °C, there is a significant increase in the emission of water molecules into the gas phase. In the temperature range 350 °C to 540 °C, a linear increase in gas release is observed.

CO_2_ is also visible, indicating the presence of carbonate phases. The thermograms of samples from solutions of chicken manure (P) and corn silage (K) are slightly different. Analysis of the DTA curve for sample P revealed as many as seven thermal effects (Figure 13).

The endothermic maxima for sample P occur at 122, 193.9, 285.5, 316.9, 846.8 and 859.8 °C, respectively, while an exothermic peak occurs at 409.5 °C. Comparison of the thermograms with the gas emission curves (Figure 14) shows that the endothermic peaks for temperature values between 30 and 430 °C are associated with the dehydration and dehydroxylation of hydrated calcium aluminates. Gas emissions from hydrate dehydration also coincide with carbon dioxide emissions, with a maximum at 415 °C. The effect of CO_2_ emissions is probably related to the oxidation of organic compounds present in the corrosive media. From a temperature of 430–445 °C, the H_2_O release curve shows a sharp decline in the dehydroxylation processes. Following this, gas emission decreases steadily until the burning of the sample is completed. In addition to the effect associated with the combustion of organic compounds, the CO_2_ release curve indicates the two-stage decarbonation of the sample (maxima at 820 °C and 850 °C). This may suggest the presence of two different phases involving carbon dioxide or characterised by the specific structure of the crystallites. A similar pattern was observed in the C_3_A subjected to biological corrosion in corn silage (K). The thermal curves (Figure 15) indicate the presence of six endothermic peaks and one exothermic peak.

Analysis of the gaseous emissions showed that the gaseous water emission starts at 50 °C and has a systematic linear increase up to a temperature of 480 °C, from which it decreases sharply and then remains almost stable until the end of the measurement. In the range from 590 °C to 620 °C, a slight increase in water emission can be observed, probably related to the dehydroxylation of the last portions of the hydration products. The carbon dioxide emission curve showed the presence of three effects—the first at 390 °C, which originates from the decomposition and oxidation of organic compounds present in the hydrated starts of synthetic C_3_A, and a double effect (Figure 16) occurring between 650 °C and 1000 °C, related to the decomposition of carbonate compounds resulting from biological processes in the test sample. The latter two CO_2_ effects may be related to the presence of different carbonate compounds or the presence of varying sizes of expanding crystallites containing bonded carbon dioxide. All the thermal effects concerning the examined samples are listed in Table 2.

### 3.4. FTIR Measurements

Figure 17 shows the surface-normalised FTIR spectra of samples of anhydrous calcium aluminate C_3_A or those treated with corrosive agents: S—not treated, anhydrous, W—subjected to water, G—subjected to pig slurry, K—treated with corn silage and with P—chicken manure.

In all spectra, four distinct groups of bands are visible (3800–3100, 1700–1200, 950–650 and below 600 cm^−1^). The first range represents mainly stretching vibrations of the OH group, while the other ranges are considered a specific fingerprint region [18]. For samples subjected to corrosive media, additional low-intensity bands in the range 3000–2500 cm^−1^ are visible (W, G, K and P). The band with a maximum at 1496 cm^−1^ seen in all samples most likely corresponds to stretching vibrations of the C-O group (νCO_3_) [18]. The absorption bands at 1464, 1080 and 873 cm^−1^ are assigned to different C-O vibration modes of carbonate groups CO_3_^2−^ [19,20]. Bands below 650 cm^−1^ probably represent vibrations of AlO_6_ octahedral groups (ca. 520 cm^−1^). A rather narrow band with a maximum at 412 cm^−1^ reflects the vibrations of the Ca-O group. The peak at 537 cm^−1^ represents bending vibrations within C-H bonds. In this band, vibrations of Si-O and Al-O are also possible [18]. An additional low-intensity band between 2800 cm^−1^ and 3200 cm^−1^ probably represents C-H group stretching modes (mostly visible in samples K and P). The band at ca. 1639 cm^−1^ can be attributed to H-O-H bonds (sample P) [21].

Additionally, differential spectra were prepared to investigate the effects of water and individual media on the C_3_A sample. The effects from the presence of corrosive media for W, G and K samples were very similar. The effect of water on the C_3_A phase (differential spectrum between hydrated and dry C_3_A phase; W *minus* S) and the most prominent effect of the biological corrosion medium (chicken manure—P) against the C_3_A sample treated with water (W) (P *minus* W) were analysed. In this way, the effect of organic matter was shown relative to the effect of the water-related hydration process.

Figure 18 shows the differential spectra between the hydrated and dry C_3_A phase or that between hydrated C_3_A and C_3_A treated with chicken manure.

## 4. Discussion

In order to recognise the biocorrosion process of C_3_A—the main component of clinker—in chicken manure (G), pig slurry (P) and corn silage (K), morphological observations of their transverse fractures were carried out using SEM, while EDS was used to analyse the chemical composition of the resulting products. For comparative purposes, analogous studies were also performed for the tricalcium aluminate/water (W) system.

Microscopic observations of hydrated samples of tricalcium aluminate (C_3_A) show a varying morphological structure depending on the type of corrosion medium used.

Analysis of the differential spectrum between the hydrated and anhydrous C_3_A phase indicated that the main effect is a band representing stretching vibrations of the OH groups, indicating the presence of higher amounts of water.

A comparison of the differential spectra from Figure 18 shows that the presence of water causes the smoothing of the bands in the 950–500 cm^−1^ range. The result is the presence of multiple maxima and minima. It is possible that this effect is due to the overlap of bands characteristic of dry and hydrated C_3_A. Interestingly, maxima appear in the spectral range 700–500 cm^−1^ and minima in the range 950–700 cm^−1^. This indicates interactions between aluminate rings and water.

Compared to samples treated with water (W), changes in the microstructure were found in practically all cases (samples G, P and K), which include the shape and size of the grains.

The corrosion process takes place in all of the applied aqueous media. Hydration of tricalcium aluminate first results in the formation of gel products, which then crystallise as a mixture of two phases: C_2_AH_8_ and C_4_AH_13_ [10]. As a result of the progressive hydration, the C_3_A grains are surrounded by a layer formed from these phases, so further hydration takes place by ion diffusion. During this chemical reaction, the C_4_AH_13_ phase is transformed into hexagonal hydrates of C_4_AH_19_, which can also be formed independently as the primary phase [10]. The end product of the tricalcium aluminate hydration process involving both of the aforementioned hexagonal phases, i.e., C_2_AH_8_ and C_4_AH_19_, is a thermodynamically stable regular hydrate of C_3_AH_6_, according to the course of the following reaction [10]:(1)$$C_{4}AH_{19}+C_{2}AH_{8}\to 2C_{3}AH_{6}+aq$$

Analysis of the morphological structure in conjunction with the X-ray analysis (Figure 8) of the tricalcium aluminate (W) showed the presence of two types of crystallite structure:−regular phase called catoite—Ca_3_Al_2_(OH)_12_;−a hexagonal one called hydrocalumite—Ca_2_Al(OH)_7_·3H_2_O.

The results of the chemical composition analyses from selected areas of the W sample indicate significant differences in the chemical composition of the two phases mentioned, which are de facto transitional reactants of the tricalcium aluminate hydration process, proceeding according to the chemical reaction in Equation (1).

Qualitative and quantitative analysis of anhydrous C_3_A using X-ray diffraction showed its high purity. Results of the analysis with the Rietveld method showed the presence of 97.5% of the cubic C_3_A phase and 2.5% of majenite (C_12_A_7_) by weight.

The C_3_A spectrum (S) is typical and shows two well-defined, dominant absorption areas. The most intensive bands appear in the area between 950–650 cm^−1^ and 500–380 cm^−1^. The maxima near 897, 863, 817, 786, 740 and 705 cm^−1^ represent AlO_4_ tetrahedral groups (stretching vibrations of Al-O bonds in aluminate rings), and those close to 520, 510, 460 and 414 cm^−1^ indicate AlO_6_ octahedral groups. The Ca-O bands appear at lower frequencies (412 cm^−1^) [18,22]. The experiments on the structure of aluminate rings in tricalcium aluminate according to X-ray data show that Al_6_O_18_ rings consist of six AlO_4_ tetrahedra connected by bridging O atoms [23]. Computation with the force field shows that the ring atoms exhibit minor variations in bond length, and the bond angles deviate slightly from the ideal tetrahedral values due to a puckered structure [24].

In the gas evolution curves, the effect of dehydration is evident at temperatures of around 50–250 °C. This fact is probably associated with the significant hygroscopicity of the synthetic tricalcium aluminate sample. Samples subjected to hydration in water and in aqueous solutions of biological media, which included pig slurry, chicken manure and corn silage, showed some similarity both in the analysis of phase composition performed by XRD methods and in the analysis of changes in thermal curves.

Samples hydrated in water (W) and in an aqueous solution of pig slurry (G) show some similarities. Sample W contains 95% of catoite Ca_3_Al_2_(OH)_12_ and 4% of hydrocalumite Ca_2_Al(OH)_7_·3H_2_O, by weight. There is also ca. 1% by weight of unreacted C_3_A. Moreover, FTIR analysis confirms the presence of water and additional peaks most probably connected to the hydrated C_3_A phase.

The analysis indicates the changes in the DTA, DTG and TG curves. First, in the temperature range from 150 °C to 340 °C, there is a gradual mass loss confirmed by endothermic effects on the DTA and DTG curves.

Initially, hexagonal crystallites of hydrated calcium aluminates undergo dehydration, followed by more stable, regular hydrate forms. A parallel analysis of gas emissions confirms a systematic increase in the H_2_O release, while negligible CO_2_ release is observed.

The low emissions come from the CO_2_ bound from the air due to the high reactivity of the C_3_A phase under high-temperature conditions. SEM microphotography and EDS spot analysis results show the presence of spherical globules, indicating the presence of regular hydrated calcium aluminates sized between 0.5 µm and 4 µm and hexagonal-shape plates with sizes ranging from 8 µm to 10 µm. The results of EDS spot analysis of the micro-areas of the sample confirm the presence of transitional forms of hydrated calcium aluminates.

The above is confirmed by the FTIR spectra analysis. In the case of samples treated with water or corrosive agents, the intensity of the bands in the range 3800–3200 cm^−1^ increases significantly, especially at wavenumbers between 3662 and 3664 cm^−1^, which is characteristic of the -OH groups (most evident for W and G samples). In addition, bands in the range 3800–3200 cm^−1^ are responsible for stretching vibrations of the OH groups—possibly also in combination with Al (Al-OH stretch). For sample S, this band should not be present [25]; however, water from the air reacts instantly with tricalcium aluminate [26]. The broadening of this band towards the lower wavenumbers indicates the presence of other O-H group vibrational modes or N-H group vibrations. It can also represent the OH states characterised by interlayer water molecules present in hydrocalumite-type material (mainly in samples K and P) or a different degree of hydrogen bond (samples W and G). The band positioned at approximately 1414 cm^−1^ is characteristic of C-O stretching vibrations [18].

In sample G treated with pig slurry, the same hydration products are present as in sample W kept in water. This includes catoite Ca_3_Al_2_(OH)_12_ and hydrocalumite phases Ca_2_Al(OH)_7_·3H_2_O, and hydrated calcium aluminate Ca_2_Al_2_O_5_·8H_2_O. Quantitative analysis of the latter phase was not performed due to the lack of a standard used for Rietvield analysis. There is 5.5% by weight of unreacted C_3_A in the sample. Differences in the obtained test results become apparent in the quantitative analysis of the tested samples. Regular hydrates—Ca_3_Al_2_(OH)_12_—are present in the amount of 61% by weight, while hexagonal hydrates are present in the amount of 15.3% by weight. The sample also contains 3% by weight of the CaCO_3_ crystalline phase, confirmed by qualitative phase composition analysis. The occurrence of calcium carbonates in samples held under conditions of biological corrosion confirms the participation of living matter in the processes of the formation of phases additional to the standard products of hydration, which are hydrated calcium aluminates. The thermal analysis confirms the results obtained during the determination of the phase composition of samples held in pig slurry. As previously mentioned, the main course of the DTA, DTG and TG curves is very similar to those obtained with samples exposed to water. Differences are shown by the occurrence of additional exothermic peaks at around 430 °C from the combustion of organic compounds present in the sample. Different from the samples subjected to water are the results of the concentration of CO_2_ emissions visualised in the gas release analysis curve, which shows a significant increase. At a temperature of around 800–870 °C, there is a doubling of the carbon dioxide peak emission, which clearly indicates the presence of carbonate compounds from reactions at the interface between living and dead matter. The doubling of the peak probably indicates the presence of the crystallites of different size derived from carbonate compounds, or slight changes in the elemental cell of calcium carbonate associated with the incorporation of ions present in the reaction medium [27].

SEM micrographs and EDS composition analysis spectra show the presence of hydrated regular calcium aluminate hydrates and hexagonal plates, indicative of the formation of a carbonate phase. Analysis of the EDS compositions of selected areas of the samples also indicates the presence of associated ions such as phosphorus, magnesium, silicon and small amounts of sulphur. C_3_A, characterised by its high solubility and high dissolution rate, reacts rapidly with S0_4_^2−^ and Ca^2+^ ions in the first instance. The occurrence of these elements confirms the possibility of their incorporation into the hydration products being formed, as well as the presence in the phase composition of secondary crystalline phases formed as a result of metabolic reactions of living matter reacting in the sample. The occurrence of an organic fraction in the phase composition is also indicated in the total composition of the quantitative analysis of the sample examined. Due to the absence of standards as well as the pseudo-crystalline nature of the organic matter, the remaining amount, i.e., 15.2 wt.%, represents the sum of all the undetermined substances in the test sample.

In anhydrous C_3_A, four atom types can be distinguished, such as calcium, aluminium, apical oxygen and ring oxygen. Upon hydration, several new atom types are generated, including oxygen and hydrogen in hydroxide ions. These atom types share similarities with AlOH groups in ettringite and with OH- ions in calcium hydroxide, respectively [24,28]. It is also possible that all the atoms participate in non-bond interaction. Al–O bonds in tetrahedral oxygen coordination can have a more covalent character and are slightly shorter than in octahedral oxygen coordination [29], while calcium is embedded in coordination environments, resulting in covalent bonding [24].

The roentgenograms, thermal curve analysis and FTIR for samples held in aqueous solutions of chicken manure (P) and corn silage (K) show many similarities. The XRD results of sample K indicate the presence of three main hydrated calcium aluminates, i.e., Ca_3_Al_2_(OH)_12_, Ca_2_Al(OH)_7_·3H_2_O and Ca_2_Al_2_O_5_·8H_2_O. The quantitative analysis of the two main components of this sample indicates 44.5%, 22.0% by weight, respectively. The quantitative content of the Ca_2_Al_2_O_5_·8H_2_O phase was not determined (no standard available). Other crystalline phases, such as CaCO_3_ and Al(OH)_3_, are also in C_3_A seasoned in an aqueous solution of corn silage. Their amount varies between 3 and 4 wt.%.

It is worth emphasising the increase in the amount of carbonate and hydroxide phases in relation to samples kept in water and in aqueous solutions of pig slurry. This can be additionally confirmed by analysing the registered thermal curves. The DTA, DTG and TG curves indicate the occurrence of seven independent thermal effects. Up to a temperature value of 318 °C, dehydration and dehydroxylation of the test sample occur. This is confirmed by the gas release analysis curve, which shows a systematic increase in H_2_O. At approximately 401 °C, there is a visible exothermic peak associated with the combustion of organic compounds in the sample. This fact is confirmed by the formation of phases in the sample and on its surface that originate from the reaction of living matter in the hydrated sample. Two overlapping peaks at 824 °C and 896 °C are visible. The endothermic effects originate from the decarbonation process confirmed by the gas emission curve of the test sample, which clearly shows two areas of emission. The differentiation of CO_2_ release from the sample into two effects probably indicates the process of carbonation of the sample due to metabolic reactions of living matter. Microphotography analysis of a sample of synthetic tricalcium aluminate subjected to an aqueous solution of corn silage indicates the presence of large regular crystallites resembling cubic catoite phases. The observed crystals are sized between 1.5 and 6.5 µm. The SEM image also shows areas of clusters of spherical crystallites indicating the presence of Al(OH)_3_ phases. The presence of calcium carbonates throughout the sample is confirmed by visible phase conglomerates.

Analysis of the EDS spectra confirms the presence of all the phases indicated above, but small amounts of phosphorus, sulphur and magnesium are present in the sample K, in addition to the main components, such as aluminium, calcium, carbon and oxygen. These elements probably build into the crystallites of the main phases, changing the shape of their elementary cells.

The XRD analysis of sample P from chicken manure shows a similarity in its diffraction pattern and thermal behaviour to that exposed in corn silage—K. The diffractograms of the samples obtained indicate the presence of crystalline phases such as regular hydrate—C_3_Al_2_(OH)_12_. The catoite phase is present in the sample in an amount of approximately 32% by weight. The second most abundant is the hydrocalumite phase Ca_2_Al(OH)_7_·3H_2_O—26.5 wt.%. Moreover, 4.5% unreacted C_3_A and other phases formed as a result of exposure conditions in aqueous solutions of biological media, which include Al(OH)_3_ and CaCO_3_, are also present. The quantitative analysis of the individual corrosion products reveals 5.1 and 12 wt.%, respectively. The formation of aluminium hydroxide and calcium carbonate depends mainly on the conditions and duration of exposure of the samples to the solutions. A change in pH towards a more acidic solution causes crystallisation and an increase in the amount of the above-mentioned phases. The band in the range 3800–3200 cm^−1^ corresponds to stretching vibrations of the OH groups—possibly also in combination with other atoms, usually metals (M), the M-OH stretch. Figure 17 and Figure 18 show significant reorganisation in this range. A negative band appears with a maximum at 3664 cm^−1^ and positive bands with maxima at 3525 cm^−1^ and 3464 cm^−1^. The XRD results are confirmed by thermal analyses. The heated sample K shows the existence of seven thermal effects, including six endothermic and one exothermic. In the temperature range from 20 °C to 317 °C, the dehydration and dehydroxylation of hydrated calcium aluminates occur in the sample. The above analysis is confirmed by the gas emission curves, where the only gas phase is water. At 409.5 °C, the DTA curve shows an exothermic peak confirming the presence of organic matter in the sample. At this point, there is an apparent increase in CO_2_ in the gas emission analysis graphs.

In addition, there are two prominent endothermic peaks in the DTG curve associated with the decomposition of carbonate compounds. The occurrence of a double heat effect (at temperatures of 846.8 °C and 895.8 °C) of the decomposition of carbonate compounds is confirmed by the curve of gas release from the heated sample. This character of decomposition indicates that the carbonate compounds should be associated with metabolic processes originating from organic matter in the sample. A lower decomposition temperature will indicate the presence of very small crystallites of newly formed carbonate compounds or a different polymorphic form of this phase. On the other hand, a higher temperature will indicate that the crystals of the carbonate phase are larger and their thermal dissociation process will take place at a slightly higher temperature.

SEM microphotograph analysis of the sample indicates the presence of spherical (cubic) crystallites most likely originating from the catoite phase. The image also shows fibre-like crystals between 0.5 µm and 6 µm in diameter, which may indicate the presence of hexagonal hydration products. Spherical crystals are also visible in the sample, indicating the presence of an aluminium hydroxide phase, which can also take the form of amorphous and shapeless clusters crystallising on the surfaces of crystallites in the sample. The results of EDS spot analysis indicate the presence of elevated carbon and calcium content in the sample. This confirms the occurrence of carbonate compounds in the investigated areas. In addition, the results of EDS analysis of fibres visible in the sample confirm the presence of calcium aluminate compounds doped with phosphorus, sulphur and magnesium in these areas.

## 5. Conclusions

The application of different corrosion agents, such as aqueous solutions of corn silage, pig slurry and chicken manure, on tricalcium aluminate affects its quantitative and qualitative composition.The following crystalline phases were determined in the phase composition of synthetic tricalcium aluminate pastes stored in water for one month: C_3_Al_2_(OH)_12_, Ca_2_Al(OH)_7_·3H_2_O and Ca_2_Al_2_O_5_·8H_2_O.Synthetic tricalcium aluminate pastes subjected to biological corrosion showed, in addition to the above-mentioned phases, the formation of secondary crystallisation products such as CaCO_3_ and Al(OH)_3_.Variable exposure conditions and possible changes in the chemical composition of the aqueous solutions of the corrosive media in which the synthetic tricalcium aluminate samples were hydrated have the effect of increasing the content of calcium carbonates and aluminium hydroxides.The increased calcium carbonate CaCO_3_ content of the samples is probably due to the presence of bacteria and the associated metabolic processes resulting from their anaerobic respiration.

## Figures and Tables

**Figure 1 materials-16-02225-f001:**
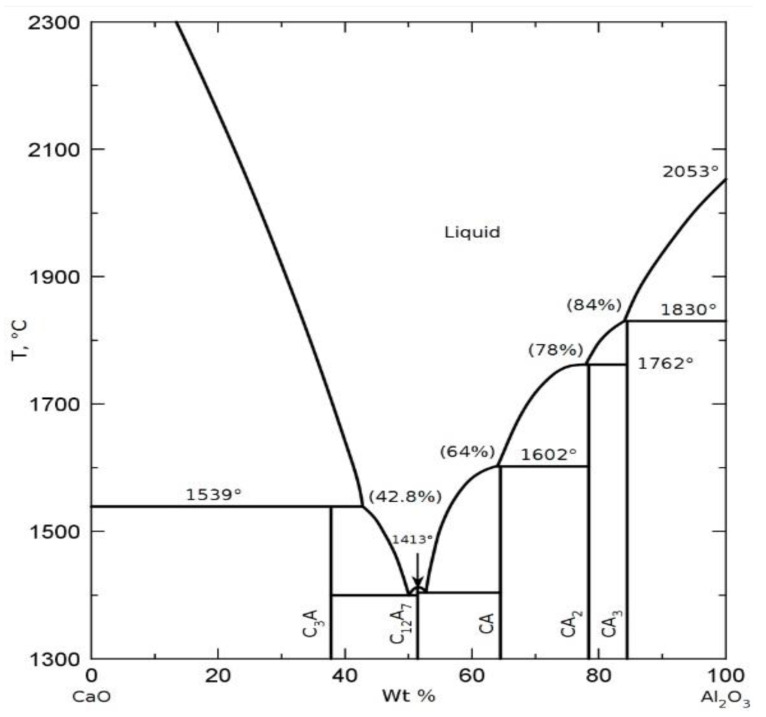
CaO-Al_2_O_3_ two component system, according to [15].

**Figure 2 materials-16-02225-f002:**
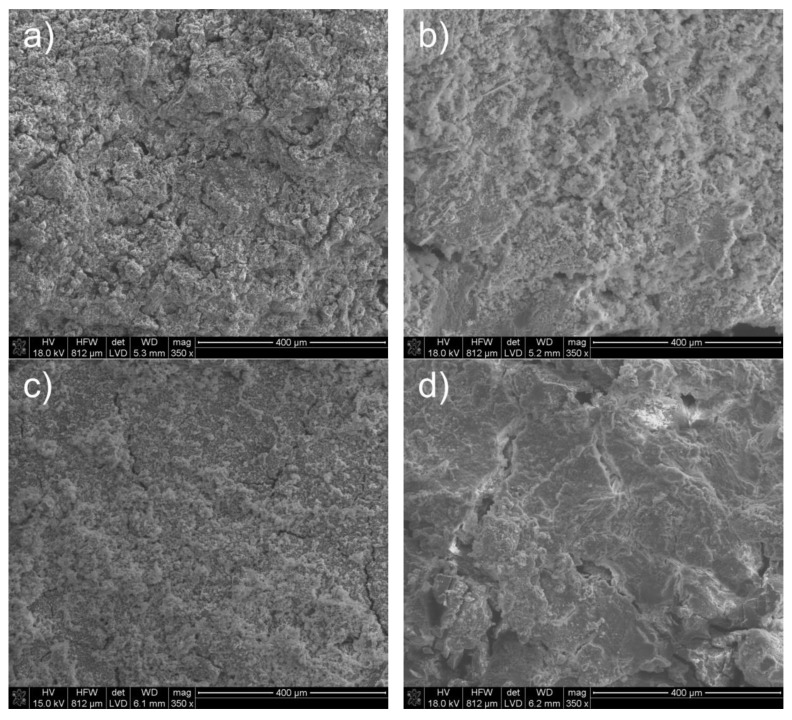
SEM microphotographs of tricalcium aluminate after exposure in (**a**) water (W); (**b**) chicken manure (P); (**c**) pig slurry (G) and (**d**) corn silage (K).

**Figure 3 materials-16-02225-f003:**
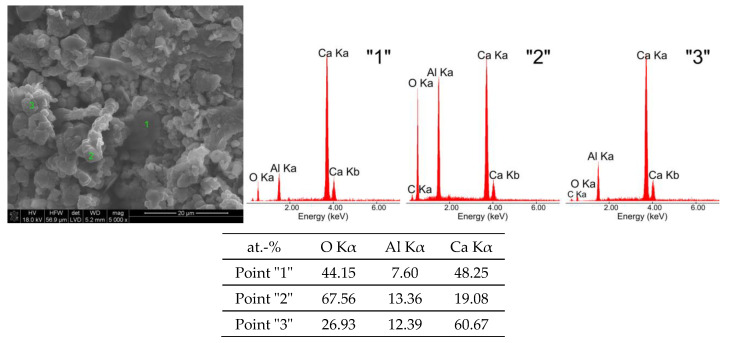
SEM microphotographs of tricalcium aluminate after exposure in water (W) and EDS quantitative point analyses of areas 1, 2, 3.

**Figure 4 materials-16-02225-f004:**
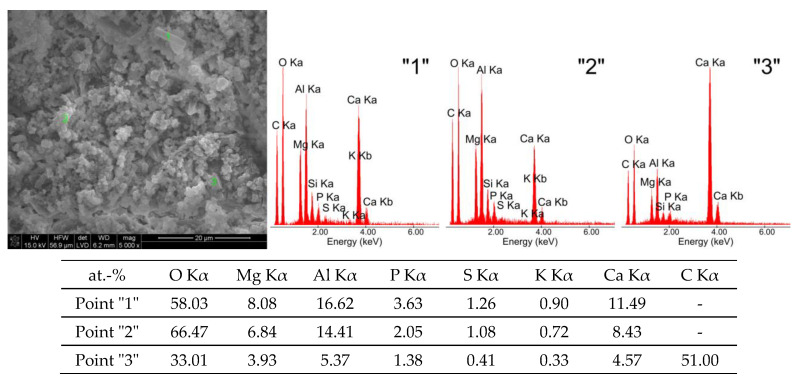
SEM microphotograph of tricalcium aluminate after exposure to pig slurry (G) and EDS quantitative point analyses of areas 1, 2, 3.

**Figure 5 materials-16-02225-f005:**
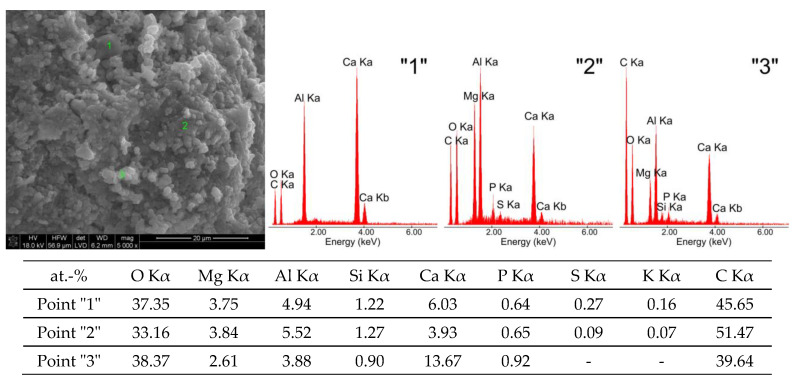
SEM microphotographs of tricalcium aluminate after exposure to corn silage (K) and EDS quantitative point analyses of areas 1, 2, 3.

**Figure 6 materials-16-02225-f006:**
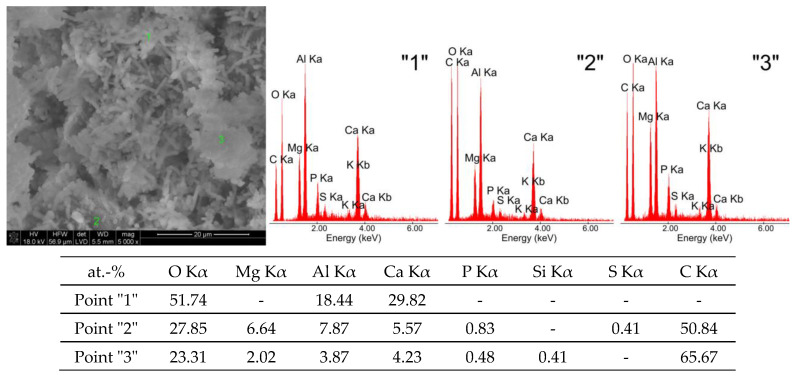
SEM microphotographs of tricalcium aluminate after exposure in chicken manure (P) and EDS quantitative point analyses of areas 1, 2, 3.

**Figure 7 materials-16-02225-f007:**
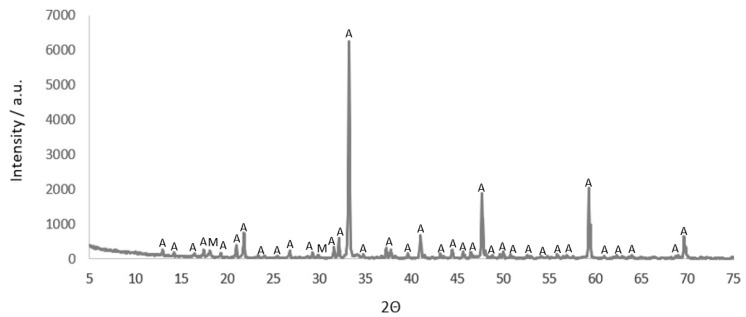
XRD spectrum of anhydrous synthetic tricalcium aluminate C_3_A (sample 1(S)). A—C_3_A; M—C_12_A_7_.

**Figure 8 materials-16-02225-f008:**
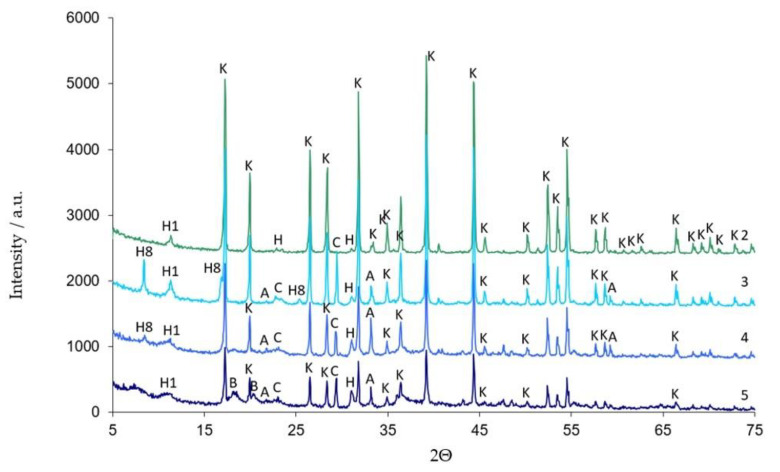
Diffractograms of the tested C_3_A samples: 2—C_3_A kept in water (W), 3—sample from pig slurry (G), 4—corn silage (K) and 5—chicken manure (5); A—C_3_A, C—CaCO_3_; H_1_—Ca_2_Al(OH)_7_·3H_2_O; H_8_—Ca_2_Al_2_O_5_·8H_2_O; K—C_3_Al_2_(OH)_12_; B—Al(OH)_3_.

**Figure 9 materials-16-02225-f009:**
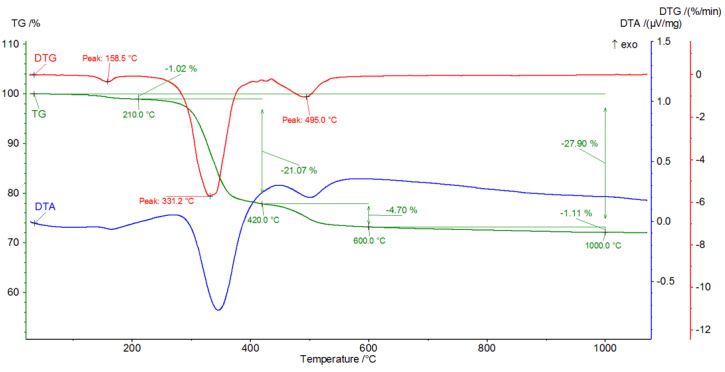
DTA/TG/DTG thermal curves of sample W as a function of temperature in the range 30–1000 °C.

**Figure 10 materials-16-02225-f010:**
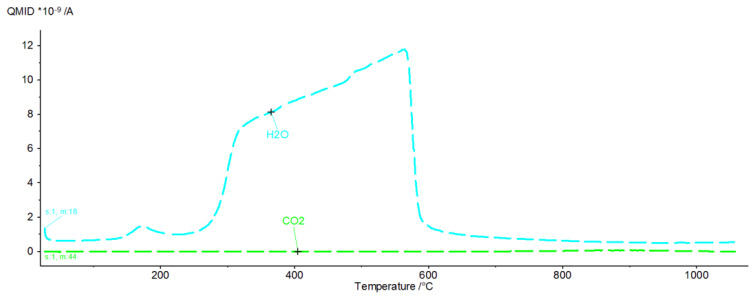
H_2_O and CO_2_ release curves from sample W as ionic current as a function of temperature.

**Figure 11 materials-16-02225-f011:**
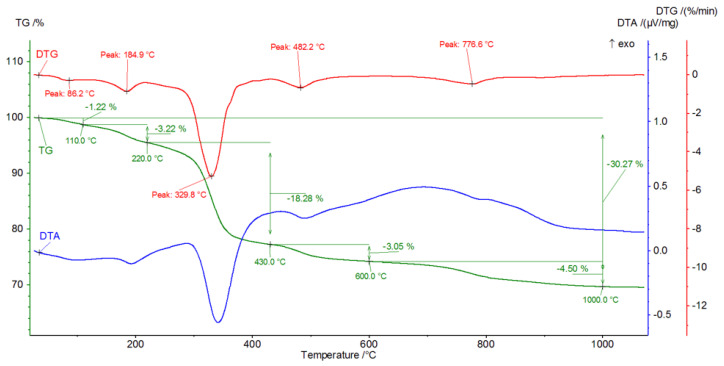
DTA/TG/DTG thermal curves of sample G as a function of temperature in the range 30–1000 °C.

**Figure 12 materials-16-02225-f012:**
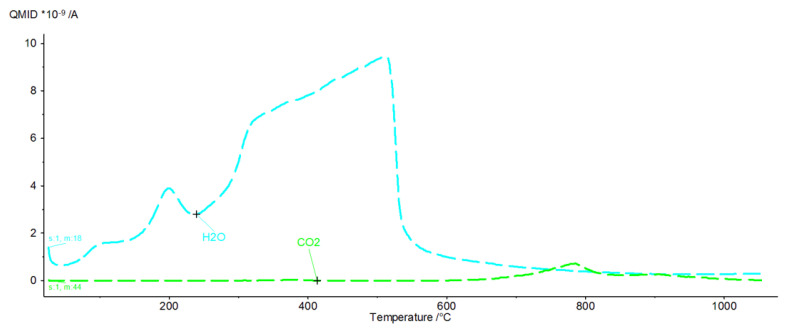
H_2_O and CO_2_ release curves from sample G as ionic current as a function of temperature.

**Figure 13 materials-16-02225-f013:**
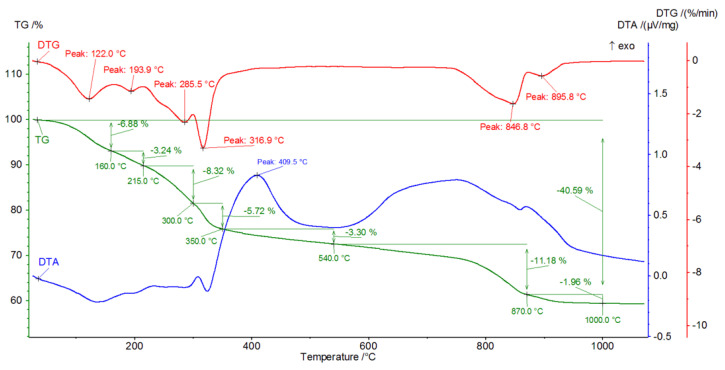
DTA/TG/DTG thermal curves of sample P as a function of temperature in the range 30–1000 °C.

**Figure 14 materials-16-02225-f014:**
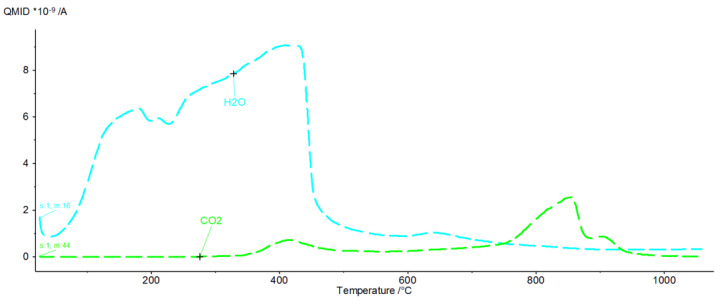
H_2_O and CO_2_ release curves from sample P as ionic current as a function of temperature.

**Figure 15 materials-16-02225-f015:**
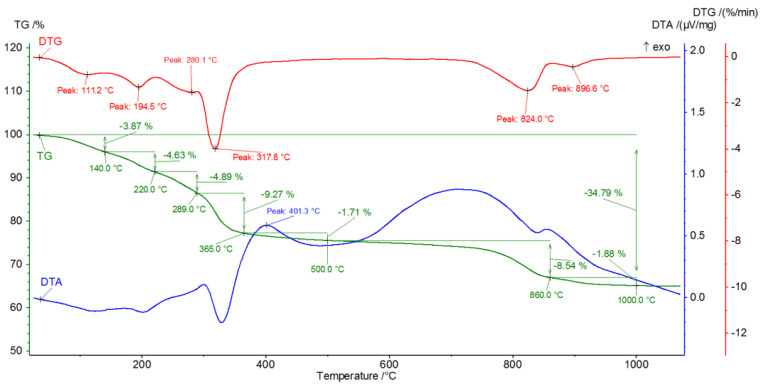
DTA/TG/DTG thermal curves of sample K as a function of temperature in the range 30–1000 °C.

**Figure 16 materials-16-02225-f016:**
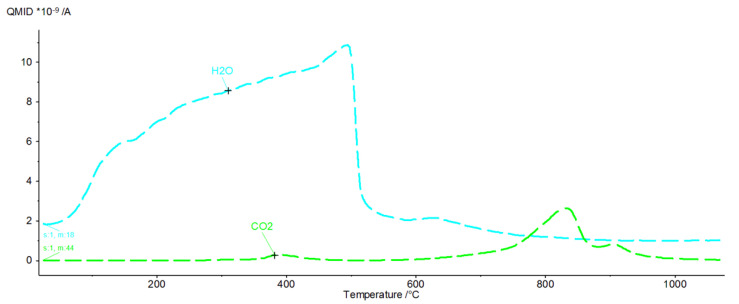
H_2_O and CO_2_ release curves from sample K as ionic current as a function of temperature.

**Figure 17 materials-16-02225-f017:**
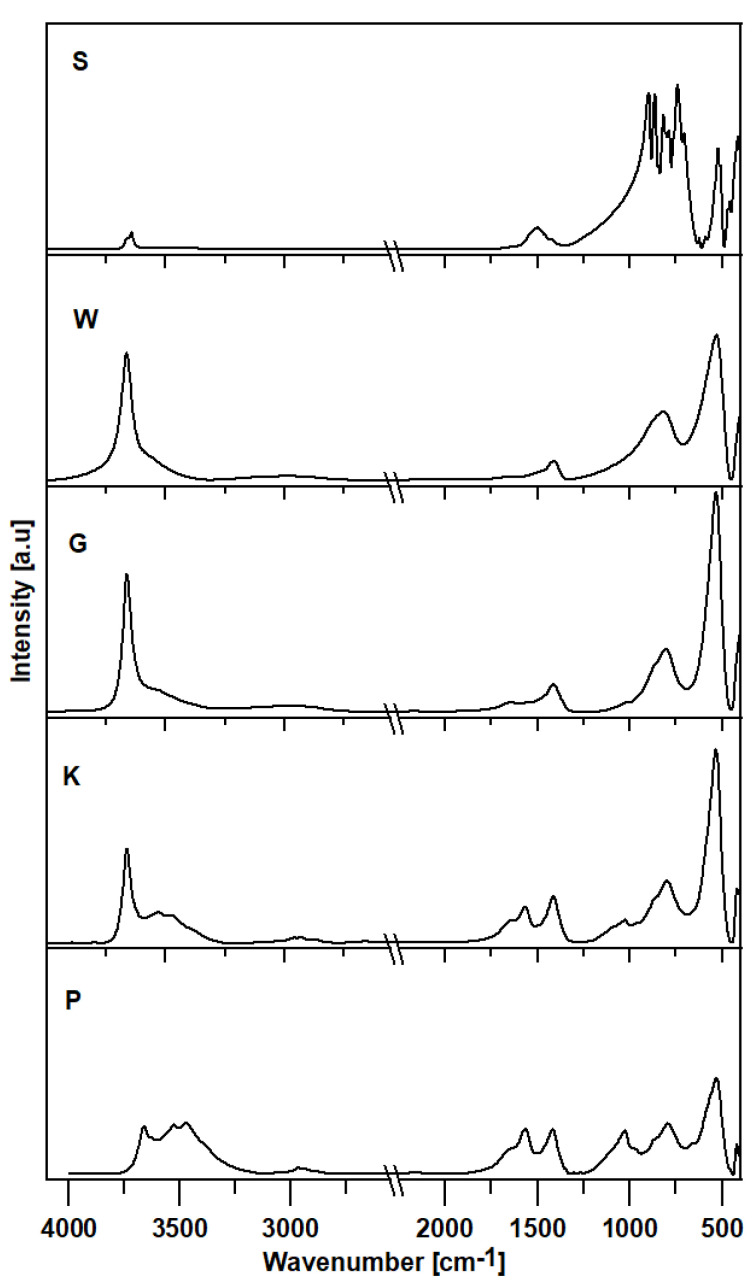
Surface-normalised FTIR spectra of calcium aluminate C_3_A (S—not treated) phase subjected to water (W), pig slurry (G), corn silage (K) or chicken manure (P) in the wavenumber range 4000–400 cm^−1^.

**Figure 18 materials-16-02225-f018:**
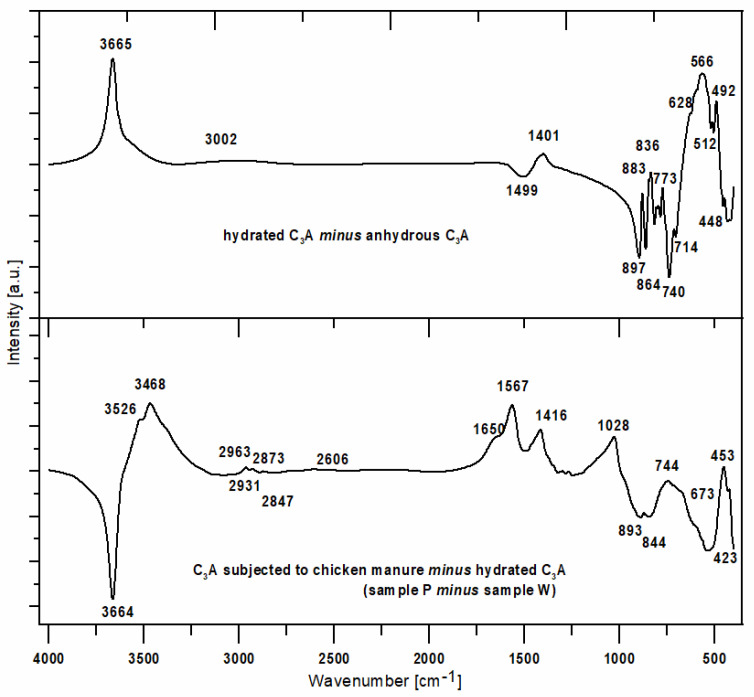
Differential spectra between the hydrated and dry C_3_A phase (hydrated C_3_A *minus* anhydrous C_3_A)—upper panel, or between hydrated C_3_A and C_3_A treated with chicken manure (sample P *minus* sample W)—lower panel. Characteristic wavenumbers, as indicated, respectively.

**Table 1 materials-16-02225-t001:** Qualitative and quantitative composition of cement slurries from C_3_A hydrated in water or corrosive media, as indicated.

Sample/Label	C_3_A	C_12_A_7_	C_3_Al_2_(OH)_12_	Ca_2_Al(OH)_7_·3H_2_O	Ca_2_Al_2_O_5_·8H_2_O	CaCO_3_	Al(OH)_3_
1 (S)—C_3_A	+(97.5) *	+(2.5)	-	-	-	-	-
2 (W)—water	+(1.0)	-	+(95.0)	+(4.0)	-	-	-
3 (G)—pig slurry	+(5.5)	-	+(61.0)	+(15.5)	+(**)	+(3.0)	-
4 (K)—corn silage	+(10.5)	-	+(44.5)	+(22.0)	+(**)	+(4.0)	+(3.2)
5 (P)—chicken manure	+(4.5)	-	+(32.0)	+(26.5)	-	+(12.0)	+(5.1)

+ phase present, - phase not present; numbers in brackets indicate weight percentage of a component, * synthetic C_3_A, anhydrous, ** no available standard for quantification analysis

**Table 2 materials-16-02225-t002:** Weight changes in C_3_A samples stored under different conditions during heating and associated thermal effects.

No.	Mass Change Δm, [%]	Temp. Range (Maximum Δm) [°C]	Type of DTA Effect	Gas Released	Type of Process
Sample W (subjected to water)					
1	−1.02	30–210 (158)	endo	H_2_O	dehydration
2	−21.70	210–420 (331)	endo	H_2_O/CO_2_	dehydration oxidation of organics
3	−4.70	420–600 (495)	endo/egzo	H_2_O/CO_2_	dehydroxylation, oxidation of organics
4	−1.11	600–1000 (-)	endo	H_2_O	carbonate decarbonation
Δm (30–1000 °C) = −27.90%					
Sample G (subjected to chicken manure)					
1	−1.22	30–110 (86)	endo	H_2_O	dehydration
2	−3.22	110–220 (185)	endo	H_2_O	dehydration
3	−18.28	220–430 (330)	endo/egzo	H_2_O/CO_2_	dehydroxylation, oxidation of organics
4	−3.05	430–600 (330)	endo	H_2_O	dehydroxylation
5	−4.50	600–1000 (776)	endo	CO_2_	carbonate decarbonation
Δm (30–1000 °C) = −30.27%					
Sample P (subjected to pig slurry)					
1	−6.88	30–160 (122)	endo	H_2_O	dehydration
2	−3.24	1640–215 (194)	endo	H_2_O	dehydration
3	−8.32	215–300 (285)	endo	H_2_O	dehydration
4	−5.72	300–350 (317)	endo	H_2_O	dehydroxylation,
5	−3.30	350–540 (-)	egzo	CO_2_	oxidation of organics
6	−11.18	540–870 (847)	endo	CO_2_	carbonate decarbonation
7	−1.96	870–1000 (896)	endo	CO_2_	carbonate decarbonation
Δm (30 °C–1000 °C) = −40.59%					
Sample K (subjected to corn silage)					
1	−3.87	30–140 (111)	endo	H_2_O	dehydration
2	−4.63	140–220 (194)	endo	H_2_O	dehydration
3	−4.89	220–289 (280)	endo	H_2_O	dehydration
4	−9.27	289–365 (318)	endo	H_2_O	dehydroxylation
5	−1.71	365–500 (-)	egzo	CO_2_	oxidation of organics
6	−8.54	500–860 (824)	endo	CO_2_	carbonate decarbonation
7	−1.88	860–1000 (896)	endo	CO_2_	carbonate decarbonation
Δm (30–1000 °C) = −34.79%					

## Data Availability

Data will be available on request.

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
