# Peer review of "The Effect of Biological Corrosion on the Hydration Processes of Synthetic Tricalcium Aluminate (C3A)"

_materials, 2023, doi:10.3390/ma16062225_

Round 1

Reviewer 1 Report

All suggestions and recommendations are presented below (or you can find them in the attached file)

The effect of biological corrosion on the hydration processes of synthetic tricalcium aluminate (C3A), a component for specialty cement used in bioenergetics

The main idea of the paper is to investigate how three corrosive environments of biological origin affect the hydration of synthetic C3A. However, the title is somehow misleading. What was author’s intension in using phrase “specialty cement” if they are analyzing synthetic C3A. It is therefore unclear to the readers why synthetic C3A was chose for the analysis. The authors selected appropriate method such as SEM-EDS, XRD, FTIR, TGA for the microstructural analysis. They found that CaCO3 and Al(OH)3 were formed in all samples cured in various environments. Despite the large amount of data presented, the discussion is weak because the authors did not disclose or indicate the main differences between samples and the significant of their work. The discussion part was too long and uninformative, as were the conclusions.

Introduction

Authors reviewed new results reported in the literature about the impact of biological attack on the hydration of cement-based materials, but selected to analyze the synthetic C3A, which accelerates corrosion in the presence of sulfate. In line 57 authors mentioned that “sulphur compounds <…> cause concrete degradation”. Therefore, the recommendation for authors is to disclose more clearly their intensions to use C3A.

Methodology

Authors selected appropriate methods for the microstructural analysis. However, the sample’s preparation method requires more explanations.

Line 132 – how the samples were compressed? In dry or wet state? What temperature and so on. What is the porosity of compressed samples? Parameters are important for readers who intend to compare results with yours.

Line 133 – the compositions of aqueous solutions or corrosive media is not presented, but they are of high important for the explanations of results.

Line 142 – where was the anhydrous C3A kept? Why do you indicate that?

Line 148 – the grinding procedure in the agate mill can cause carbonation, cannot it?

Line 148 – Can you select to which tests such a grinded samples were subjected?

Line 157 – What is the difference between two grinding procedures? It recalls with line 148.

Line 177 – Are the grains of 0.10 mm acceptable for the TGA measurements? For X-ray you used 1 g of powder sieved trough 0.01mm sieve, while 0.075 mg of material with grains of 0.1 mm in size you selected for TGA?

Line 189 – more details about sample’s grain size and mass are needed.

Results

Line 196 – recommendation for the future – WD in all images should be the same.

Line 201 – EDS spectra alone does not give information about the quantity of the elements. The table with the amount of elements should be presented.

Line 216, 219, 229,232, 242  - The point analysis is not acceptable for the detection of crystalline (C3AH, catoite, Al23*H2O phases). XRD data should be presented before SEM but not after. Moreover, the selection of elemental mapping to point analysis would be more informative and less misleading.

Line 253 – what is the composition of pig, corn silage and chicken slurries?

Line 265 – why XRD pattern of anhydrous C3A is not inserted in Fig.8? Is it so difficult?

Line 266 – why the peaks of crystalline phases are not indicated in the figure?

Line296 – Where is the anhydrous C3A figure? Where is Figure S1 and S2?

Line 304-307 – what are you talking about? What is the idea of those sentences?

Line 307 – what is it - “maize”?

Line 310 Figure 9 and all others – why all these curves were not putted into one Figure as it was done with XRD data?

Line 314 and all other TGA Figures – there are lot of repetitions in your text. You summarized weight losses of each sample so why do you repeat them in the text. With great respect to YOU, the text is boring for the reader and uninformative. You should indicate the differences between samples and give the explanation why these differences occurred.

Line 354 – what is the difference between dehydration and dihydroxylation? If you indicated that it must be important.

Line 361 – is the term “decarbonylation” acceptable? What is the difference between decarbonation and decarbonylation? Thank you for the explanations in advance.

Line 421 – Why Fig. 18-19 was not inserted into one like you did in Fig.17? JPEG images look very unprofessional. You can divide the spectra and present two or three smaller figures instead of one.

Discussion

Line 439 – what do you expected to be? Is the presence of water not logical?

Line 518-523 (line 564-567)– You made conclusions based only on SEM-EDS spot analysis (!?). I strongly suggest you to revise these sentences involving XRD or other data.

Line 545 – The pH data are missing. Insert pH measurement data, please, otherwise delete this sentence.

Line 427-664 – the discussion section is too long. You are describing the figures without the real discussion on what is new or significant and how your obtained results contribute to those presented by others. It is your chose to revise this section or not; however, I suggest you to think about the reader who you want to communicate – will he be interested in your results if you are just describing figures? Will your paper appear in somebody’s reference list except yours?

Conclusions

The main question is – does the nature of biological environmental has an impact on the hydration of C3A?

Line 667 – what corrosive solution did you use?

Line 676 – is the content important? If it is, so indicate the percentage or values.

Line 680 – How about Al(OH)3? Why it appeared and increased?

Line 683 – It would be nice if you describe more about the effect of P, S, Mg, Si on hydration of C3A in results and discussion sections. If you do not intend to do that, then conclusion Nr.6 is unacceptable for the work you did.

Author Response

Answers to Comments by Reviewer 1:

Answer to Reviewer comments on the manuscript entitled: "The effect of biological corrosion on the hydration processes of synthetic tricalcium aluminate (C3A), a component for specialty cement used in bioenergetics"

(Manuscript ID materials - 2234811R1)

            The authors would like to thank the Reviewer for valuable suggestions and comments, which helped to improve the quality of our manuscript. We have carefully gone through the all the comments and introduced the suggested changes. They are listed in our point-by-point response to the Reviewer’s comments below.

Remark 1: The main idea of the paper is to investigate how three corrosive environments of biological origin affect the hydration of synthetic C3A. However, the title is somehow misleading. What was author’s intension in using phrase "specialty cement" if they are analyzing synthetic C3A. It is therefore unclear to the readers why synthetic C3A was chose for the analysis. The authors selected appropriate method such as SEM-EDS, XRD, FTIR, TGA for the microstructural analysis. They found that CaCO3 and Al(OH)3 were formed in all samples cured in various environments. Despite the large amount of data presented, the discussion is weak because the authors did not disclose or indicate the main differences between samples and the significant of their work. The discussion part was too long and uninformative, as were the conclusions.

Response: We thank you very much for this comment. We would like to note that due to the small number of published papers in the field of corrosion of the C3A phase in various biological media, the present study is of novel character. A detailed discussion, supported by conclusions of a scientific and applied nature, will be possible once a full series of articles on this topic has been published.

Remark 2: Introduction: Authors reviewed new results reported in the literature about the impact of biological attack on the hydration of cement-based materials, but selected to analyze the synthetic C3A, which accelerates corrosion in the presence of sulfate. In line 57 authors mentioned that "sulphur compounds <…> cause concrete degradation". Therefore, the recommendation for authors is to disclose more clearly their intensions to use C3A.

Response: Thank you for this comment. The study of the C3A phase in terms of its resistance to biological corrosion in the media used in this work is the first in a series of planned studies dedicated to this topic. Subsequently, other clinker phases such as C4AF, C3S, C2S, CaSO4x2H2O and CaO will be investigated. After analysing the clinker phases and other oxides present in their composition, a model cement will be developed, which will also be subjected to the influence of biological corrosion in terms of its durability. Unfortunately, we could not disclose the entire project in the publication for concern that the idea might be stolen. Therefore we also decided to shorten the title just to: The effect of biological corrosion on the hydration processes of synthetic tricalcium aluminate (C3A).

Remark 3: Methodology: Authors selected appropriate methods for the microstructural analysis. However, the sample’s preparation method requires more explanations. Line 132 – how the samples were compressed? In dry or wet state? What temperature and so on. What is the porosity of compressed samples? Parameters are important for readers who intend to compare results with yours.

Response: Thank you for this comment. The reviewer's suggested corrections have been included in the manuscript. We did not measure the samples porosity and as all samples were prepared the same way we were able to compare the effects.

Remark 4: Line 133 – the compositions of aqueous solutions or corrosive media is not presented, but they are of high important for the explanations of results.

Response: Thank you very much for this remark. Chemical analysis of corrosive media such as pig slurry has been given in our earlier publications. In the course of the work, the authors assumed that the key issue in terms of assessing the effect of the type of media used on the corrosion of the bonding materials is the change in pH of the media in which the slurries are exposed. Such data can be found in a smaller article. Nevertheless, the results of the analyses of the chemical composition of the biological media will be included in future papers.

Remark 5: Line 142 – where was the anhydrous C3A kept? Why do you indicate that?

Response: Thank you for this comment. The reviewer's suggested corrections have been included in the manuscript. Over the 28-day experimental period, anhydrous sample was kept in a desiccator.

Remark 6: Line 148 – the grinding procedure in the agate mill can cause carbonation, cannot it?

Response: We thank you for this comment. We agree that the milling process can induce carbonation. Taking this into account, all samples were milled according to the same procedures, so it can be assumed that these samples achieved an analogous degree of carbonatization. With this assumption, there is a reasonable basis for comparing the corrosion progress of the tested materials in the aggressive media used.

Remark 7: Line 148 – Can you select to which tests such a grinded samples were subjected?

Response: Thank you for this question. The ground samples were subjected to XRD, SEM, EDS, FTIR, DTA, TG and DTG studies.

Remark 8: Line 157 – What is the difference between two grinding procedures? It recalls with line 148.

Response: There is no difference in the grinding procedure of the tested samples. Part of the manuscript text concerning grinding process has been unified.

Remark 9: Line 177 – Are the grains of 0.10 mm acceptable for the TGA measurements? For X-ray you used 1 g of powder sieved trough 0.01 mm sieve, while 0.075 mg of material with grains of 0.1 mm in size you selected for TGA?

Response: Thank you very much for this specific query. Of course, we ground the samples to a grain size of 0.01 mm when examining them using both TGA and X-ray techniques. Appropriate corrections have been included in the manuscript.

Remark 10: Line 189 – more details about sample’s grain size and mass are needed.

Response: Thank you for this comment. The grain size of the test samples for each of the test methods used was at the same level, namely 0.01 mm.

Remark 11: Results: Line 196 – recommendation for the future – WD in all images should be the same.

Response: Thank you for this comment. In the next stage of the research work, the above suggestion will be taken into account in the editing of the manuscripts.

Remark 12: Line 201 – EDS spectra alone does not give information about the quantity of the elements. The table with the amount of elements should be presented.

Response: We agree with the reviewer. Quantitative chemical compositions from selected areas of the samples were examined using spot EDS analysis The results are summarised in the relevant tables.

Remark 13: Line 216, 219, 229,232, 242  - The point analysis is not acceptable for the detection of crystalline (C3AH, catoite, Al2O3´H2O phases). XRD data should be presented before SEM but not after. Moreover, the selection of elemental mapping to point analysis would be more informative and less misleading.

Response: It was not the intention of the authors of this paper to identify the exact phase composition of the samples studied based on the results of EDS spot analyses. On the basis of the morphology of the crystallites in conjunction with the results of the X-ray analyses and the chemical composition of the EDS, it was only attempted to establish, to a greater approximation, the phase composition of the crystallites visible on the SEM microphotographs.

Remark 14: Line 253 – what is the composition of pig, corn silage and chicken slurries?

Response: The chemical analysis of pig slurry has been the subject of previous studies: Sujak, A.; Pyzalski, M.; Durczak, K.; Brylewski, T.; Murzyn, P.; Pilarski, K. Studies on Cement PastesExposed to Water and Solutions of Biological Waste. Materials 2022, 15, 1931. https:// doi.org/10.3390/ma15051931

Remark 15: Line 265 – why XRD pattern of anhydrous C3A is not inserted in Fig. 8? Is it so difficult?

Response: The authors attempted to include XRD results for anhydrous C3A in the bulk X-ray diffractogram; however, the intensity of reflections from anhydrous phases compared to those for hydrated samples differed too much. Due to the lack of readability of such a compilation, this idea was abandoned.

Remark 16: Line 266 – why the peaks of crystalline phases are not indicated in the figure?

Response: Thank you for your remark; the corrected figure has been included in the manuscript.

Remark 17: Line 296 – Where is the anhydrous C3A figure? Where is Figure S1 and S2?

Response: The TGA results on anhydrous C3A (Figure S1 and S2) are included in the supplement of this manuscript.

Remark 18: Line 304-307 – what are you talking about? What is the idea of those sentences?

Response: The authors of this paper grouped the TGA results, using the criterion of the similarity of the thermal curves.

Remark 19: Line 307 – what is it – "maize"?

Response: We thank the reviewer for this remark. Word "maize" has been replaced with word "corn".

Remark 20: Line 310 Figure 9 and all others – why all these curves were not putted into one Figure as it was done with XRD data?

Response: In the opinion of the authors, placing the curves on a summary chart would have made the analysis of the research results obtained much more difficult.

Remark 21: Line 314 and all other TGA Figures – there are lot of repetitions in your text. You summarized weight losses of each sample so why do you repeat them in the text. With great respect to YOU, the text is boring for the reader and uninformative. You should indicate the differences between samples and give the explanation why these differences occurred.

Response: Thank you for this comment. The study of the C3A phase in various corrosion media represents a section of the total work planned by the authors. Therefore, the authors have chosen to provide a detailed description of the studies performed in this manuscript. This valuable suggestion by the reviewer will be implementable once the comprehensive physico-chemical studies envisaged in the execution of this research topic have been performed.

Remark 22: Line 354 – what is the difference between dehydration and dihydroxylation? If you indicated that it must be important.

Response: Yes, these are two different chemical processes. We would like to point out that dehydration refers to the removal of moisture from the test sample, while dehydroxylation is related to the detachment of hydroxyl groups.

Remark 23: Line 361 – is the term "decarbonylation" acceptable? What is the difference between decarbonation and decarbonylation? Thank you for the explanations in advance.

Response: In this case, there was an error; it should be 'decarbonation', which was included in the manuscript.

Remark 24: Line 421 – Why Fig. 18-19 was not inserted into one like you did in Fig. 17? JPEG images look very unprofessional. You can divide the spectra and present two or three smaller figures instead of one.

Response: Fig 17 is basically illustrative. We felt that such a picture clearly shows the main changes occurring in the samples. We agree with the reviewer that it would be better to combine Figs 18 and 19. We have made this correction.

Remark 25: Discussion: Line 439 – what do you expected to be? Is the presence of water not logical?

Response: Yes, it is logical. This particular sentence concerns the analysis of the differential spectrum between hydrated and anhydrous sample. FTIR technique is very sensitive to the presence of water. Since the surface water was removed the only water present in the sample is that interlayer one. This proves that the water is bonded internally. Therefore increase in the intensity of a FTIR spectra in the region characteristic of OH stretching vibration indicates more bonded water molecules.

Remark 26: Line 518-523 (line 564-567)– You made conclusions based only on SEM-EDS spot analysis (!?). I strongly suggest you to revise these sentences involving XRD or other data.

Response: Thank you for this comment. All conclusions at the end of the paper are based on the findings obtained using all research methods. Some findings were based on selected techniques, as in this case.

Remark 27: Line 545 – The pH data are missing. Insert pH measurement data, please, otherwise delete this sentence.

Response: The indicated sentence has been removed

Remark 28: Line 427-664 – the discussion section is too long. You are describing the figures without the real discussion on what is new or significant and how your obtained results contribute to those presented by others. It is your chose to revise this section or not; however, I suggest you to think about the reader who you want to communicate – will he be interested in your results if you are just describing figures? Will your paper appear in somebody’s reference list except yours?

Response: We greatly agree with this comment. To date, few papers have been published on the corrosion of C3A under exposure to biological media. We are unable to refer to studies from other groups at this time. The reviewer's comments will be taken into account when editing future papers on this topic and as other authors' publications on this topic become available. With regard to shortening the text for the sake of the reader we have tried to do so where we could.

Remark 29: Conclusions: The main question is – does the nature of biological environmental has an impact on the hydration of C3A?

Response: We thank you for this question. There is no doubt that the nature of the corrosive environment used has a significant effect on the hydration and phase composition of the tested samples.

Remark 30: Line 667 – what corrosive solution did you use?

Response: Medium korozyjne wskazano w sekcji Materials and Methods: corn silage, pig slurry and chicken manure

Remark 31: Line 676 – is the content important? If it is, so indicate the percentage or values.

Response: Thank you for this question. The conclusion is of a general nature. The manuscript provides a comprehensive answer to this question.

Remark 32: Line 680 – How about Al(OH)3? Why it appeared and increased?

Response: Thank you for this question. The conclusion is of a general nature. The manuscript presents the answer to this question.

Remark 33: Line 683 – It would be nice if you describe more about the effect of P, S, Mg, Si on hydration of C3A in results and discussion sections. If you do not intend to do that, then conclusion Nr.6 is unacceptable for the work you did.

Response: Thank you for this suggestion. Indeed, the indicated conclusion was formulated 'over the top'. This conclusion has been removed from the manuscript.

Thank you again for the opportunity to improve the quality of our manuscript. We hope that we have answered the Reviewer's remarks in a satisfactory manner.

                                                                                               Kind regards

                                                                                               Authors

Reviewer 2 Report

In the article, physical chemical studies of the structure and phase composition of the products of hydration of tricalcium aluminate in various biologically active media are carried out. The results of these studies are of certain scientific and practical interest and can be used to predict possible biological corrosion processes in structures using Portland cement.

Notes:

1. In order to understand the features and mechanism of the process of hydration and hardening of tricalcium aluminate in the three accepted aggressive media - pig manure, corn silage and chicken manure, as well as to make the necessary generalizations, it would be advisable to bring their chemical and physico-chemical parameters.

2. The article does not provide a theoretical analysis of the mechanism of the possible destructive influence of the studied biologically active media, depending on their composition and characteristics.

3. It would be important to show how the physical and mechanical properties of the aluminate phase and Portland cement change with its different content and exposure to the corrosive media used.

4. As is known, in Portland cement, the aluminate phase during its hydration in the presence of gypsum additive passes into hydrosulfaluminate, the composition of which depends on the composition of the cement, it is of interest to reveal the features of the influence of biologically active media on the formed hydrosulfoaluminates.

5. To develop special recommendations on the composition of cement and concrete that are resistant to biological corrosion, the authors should continue research in cement systems with different contents of tricalcium aluminate and other minerals, on the basis of which necessary generalizations and practical conclusions can be drawn.

Author Response

Answers to Comments by Reviewer 2:

Answer to Reviewer comments on the manuscript entitled: "The effect of biological corrosion on the hydration processes of synthetic tricalcium aluminate (C3A), a component for specialty cement used in bioenergetics"

(Manuscript ID materials - 2234811R1)

            The authors would like to thank the Reviewer for valuable suggestions and comments, which helped improve the quality of our manuscript. We have carefully gone through the all the comments and introduced the suggested changes. They are listed in our point-by-point response to the Reviewer’s comments below.

Remark 1: In the article, physical chemical studies of the structure and phase composition of the products of hydration of tricalcium aluminate in various biologically active media are carried out. The results of these studies are of certain scientific and practical interest and can be used to predict possible biological corrosion processes in structures using Portland cement.

In order to understand the features and mechanism of the process of hydration and hardening of tricalcium aluminate in the three accepted aggressive media – pig manure, corn silage and chicken manure, as well as to make the necessary generalizations, it would be advisable to bring their chemical and physico-chemical parameters.

Response: Many thanks for this comment. The authors assumed that a key parameter in assessing the influence of corrosion media on the corrosion of the samples tested would be knowledge of the pH changes of these biological media. Such data can be found in this article. The chemical analysis of corrosive media, such as pig slurry and chicken manure, was the subject of our previous work on preliminary studies in this research topic. Analysis of the chemical composition of the aforementioned corrosive media will be included in future work.

Remark 2:  The article does not provide a theoretical analysis of the mechanism of the possible destructive influence of the studied biologically active media, depending on their composition and characteristics.

Response: Thank you for this relevant comment. Of course, there are known mechanisms for the influence of bacteria on the processes and chemical reactions occurring at the cement/bacteria interface. However, in the current series of research papers devoted to the analysis of the influence of biological media on the corrosion of clinker phases, the authors have refrained from a detailed description of the corrosion reaction mechanisms. The focus has been on the larger effects. Conclusions including the aforementioned aspects will be drawn at a later stage of the work devoted to this research topic.

Remark 3:  It would be important to show how the physical and mechanical properties of the aluminate phase and Portland cement change with its different content and exposure to the corrosive media used.

Response: We would like to thank you very much for this comment. This article is one of an initial series of papers on the biological corrosion of clinker phases, which have been directed at determining the effects of the biological media used on the chemical and phase composition and microstructure of the samples studied. The focus was on spectroscopic analysis techniques. The research thus planned will be followed by the preparation of a model cement resistant to the aggressive influence of the biological media used. Mechanical strength tests of the prepared mixtures are planned.

Remark 4: As is known, in Portland cement, the aluminate phase during its hydration in the presence of gypsum additive passes in to hydro sulfaluminate, the composition of which depends on the composition of the cement, itis of interest to reveal the features of the influence of biologically active media on the formed hydrosulfoaluminates.

Response: Many thanks for this interesting suggestion. In this paper, the authors focused on determining the influence of corrosive environments on the phase composition and microstructure of pastes made from the C3A phase. Given the limited state of knowledge achieved in this research topic, the authors decided to first investigate the influence of corrosive media on the reactivity of the 'pure' C3A phase. The aim of this approach was to possibly exclude the formation of ettringite phases or derivatives of this phase in reaction with the media used. The study of the stability of the ettringite phase in the environment of the applied corrosion media will be the subject of another research work.

Remark 5:  To develop special recommendations on the composition of cement and concrete that are resistant to biological corrosion, the authors should continue research in cement systems with different contents of tricalcium aluminate and other minerals, on the basis of which necessary generalizations and practical conclusions can be drawn.

Response: We agree with the reviewer. The authors plan to continue the research related to the present topic. The work is directed towards the design of a cement composition and, consequently, concrete fully resistant to the effects of biological corrosion, particularly in the bioenergy industry, i.e. in biogas plants.

Thank you again for the opportunity to improve the quality of our manuscript. We hope that we have answered the Reviewer's remarks in a satisfactory manner.

                                                                                               Kind regards

                                                                                               Authors

Reviewer 3 Report

The subject is interesting and represents a good piece of work; however, the authors should improve and streamline the abstract to make it easier to read. They should avoid the use of long sentences and should also support any claimed information with references throughout the manuscript.

The following will help improve the quality of the work.

L14: I believe that changing "the biological corrosion" to "the biological deterioration or degradation" is more scientific.

L15-L16: It is a pure curing process! “Over a period of one month, synthetic C3A was subjected to corrosion in maize silage, pig slurry and chicken manure while corrosion in water was used as a reference.” Please rephrase this sentence based on curing, not corrosion!

L17: “microstructure, and structure”. Actually, it is microstructure, as you are using these techniques.

L22: “formed by hydration” Hydration of what?

L24: “in the reacted preparations” What do you mean by "reacted preparation"? Please revise!

L26-L28: “The crystalline phases formed as a result of secondary crystallisation represent biological corrosion products, probably resulting from the reaction of hydrates with living matter associated with the presence of bacteria in the reaction medium.” Actually, this sentence is very confusing! What do you mean by living matter? Do you mean biomass?

The abstract in its current form seems as if it discusses two different things. The flow of the main idea of the work is interrupted in many places. Please revise the whole abstract and make it more readable and smooth.

L34: “biological corrosion” I still believe it is a biological degradation, as confirmed in L36: "The progressive degradation."

L39-L40: In Los Angeles, about 10% of sewer pipes are damaged by corrosion and the cost of restoration reaches $400 million. Please add a reference!! Corrosion here refers to steel in concrete.

L44: “reduced to” I know it is a chemical oxidation/reduction reaction, but it is better to use "converted to"

L54-L57: too long sentence that should be shortened or split and supported by a reference!

L106-L107: Preparation of the raw materials included homogenisation of the raw material portions (1-3 kg). Please specify in detail the amount of each proportion, the fineness, the mixing speed, etc. This (1-3 kg) is not clear!

L116: what do you mean by “the occurrence of secondary segregation of the ingredients”?

L128: what is the issue of “the formation of mayenite”? be more definitive please! Let readers understand the issue of mayenite formation !

L129-L130: Please provide more detail for the following sentence: “The sintered samples were then cooled in air resulting in a cubic C3A phase (see Fig. 7 below).

L133-L134: Are they diluted or following a special method of preparation? Please provide details, such as the sample-to-curing medium ratio, the chemical composition of these media, etc.

L186: an FTIR spectrometer

L193: at a magnification of 350x

L207: Exposure of tricalcium aluminate to a pig slurry, the preparation of slurry should be described.

Author Response

Answers to Comments by Reviewer 3:

Answer to Reviewer comments on the manuscript entitled:"The effect of biological corrosion on the hydration processes of synthetic tricalcium aluminate (C3A), a component for specialty cement used in bioenergetics"

(Manuscript ID materials - 2234811R1)

            The authors would like to thank the Reviewer for valuable suggestions and comments, which helped improve the quality of our manuscript. We have carefully gone through the all the comments and introduced the suggested changes. They are listed in our point-by-point response to the Reviewer’s comments below.

Remark 1: The subject is interesting and represents a good piece of work; however, the authors should improve and streamline the abstract to make it easier to read. They should avoid the use of long sentences and should also support any claimed information with references throughout the manuscript. The following will help improve the quality of the work. L14: I believe that changing "the biological corrosion" to "the biological deterioration or degradation" is more scientific.

Response: Thank you very much for these comments. The sentence from L14 has been reworded as suggested by the Reviewer. Over a period of one month, a sample of synthetic C3A was subjected to hydration, hardening and at the same time corrosion processes in maize silage, pig slurry and chicken manure. Samples subjected to the above processes in distilled water were used as a reference system. We also changed the term "the biological corrosion" into "the biological degradation".

Remark 2: L15-L16: It is a pure curing process! "Over a period of one month, synthetic C3A was subjected to corrosion in maize silage, pig slurry and chicken manure while corrosion in water was used as a reference." Please rephrase this sentence based on curing, not corrosion!

Response: We thank very much for this relevant comment. The reviewer's suggestion has been incorporated into the text of the manuscript. "During one month of setting and hardening, synthetic C3A was subjected to corrosion in corn silage, pig slurry and chicken manure. The hardening process of the C3A phase in water was taken as a reference."

Remark 3: L17: "microstructure, and structure". Actually, it is microstructure, as you are using these techniques.

Response: Thank you for this comment. The proposed amendment has been included in the manuscript.

Remark 4: L22: "formed by hydration" Hydration of what?

Response: The main crystalline phases formed by hydration of the examined samples in water as well as in corrosive media are the catoite (Ca3Al2(OH)12) and hydrocalumite (Ca2Al(OH)7×3H2O) phases. This has been corrected.

Remark 5: L24: "in the reacted preparations" What do you mean by "reacted preparation"? Please revise!

Response: We meant „in hydrating samples”.

Remark 6: L26-L28: "The crystalline phases formed as a result of secondary crystallization represent biological corrosion products, probably resulting from the reaction of hydrates with living matter associated with the presence of bacteria in the reaction medium." Actually, this sentence is very confusing! What do you mean by living matter? Do you mean biomass?

Response: Thank you for this comment. In the composition of corrosive media, due to their origin, there are bacteria (living matter) which, in connection with the highly reactive hydration products (dead matter) of the C3A phase, react to form secondary products resulting from the metabolic processes of anaerobic bacterial respiration. Chicken manure, corn silage and pig slurry are part of the biomass used during the methane reaction in biogas plants.

Remark 7: The abstract in its current form seems as if it discusses two different things. The flow of the main idea of the work is interrupted in many places. Please revise the whole abstract and make it more readable and smooth.

Response: The executive Abstract has been thoroughly re-worded.

Remark 8: L34: "biological corrosion" I still believe it is a biological degradation, as confirmed in L36: "The progressive degradation."

Response: Thank you for this comment, we have corrected the text in the indicated line of the manuscript.

Remark 9: L39-L40: In Los Angeles, about 10% of sewer pipes are damaged by corrosion and the cost of restoration reaches $400 million. Please add a reference!! Corrosion here refers to steel in concrete.

Response: Thank you for this comment. The relevant literature reference has been added to the manuscript.

Remark 10: L44: "reduced to" I know it is a chemical oxidation/reduction reaction, but it is better to use "converted to".

Response: The correction suggested by the reviewer to the content of the manuscript has been done.

Remark 11: L54-L57: too long sentence that should be shortened or split and supported by a reference!.

Response: The correction suggested by the reviewer to the content of the manuscript has been done.

Remark 12: L106-L107: Preparation of the raw materials included homogenisation of the raw material portions (1-3 kg). Please specify in detail the amount of each proportion, the fineness, the mixing speed, etc. This (1-3 kg) is not clear!.

Response: Thank you for this comment. Data on the fineness of the starting raw materials is detailed in the standards related to the purity of the raw materials used. In the present experiment, raw materials of "purity for analysis" were used. The degree of fineness was adequate to be able to determine the indicated purity of the raw material (according to the product sheet). 1-3 kg refers to raw materials mixed to the raw material bearing. More precisely, it will be according to stoichiometric ratios as described later in this paper. Actually the process was mixing of the material and preventive homogenisation. Deval drum rotated with a speed of 40 rpm. This information has been added to manuscript text.

Remark 13: L116: what do you mean by "the occurrence of secondary segregation of the ingredients"?.

Response: We thank you for this comment. The secondary segregation of components into raw material bearing is related to the specific surface area and shape of the calcium carbonate and alumina grains that were used for raw material bearing. If a long homogenisation process of the ingredients is planned without the use of rubber balls, this phenomenon of secondary segregation can occur.

Remark 14: L128: what is the issue of "the formation of mayenite"? be more definitive please! Let readers understand the issue of mayenite formation !.

Response: Thank you for this comment. The mayenite phase is thermodynamically the most stable phase in the CaO-Al2O3 system. In the situation of errors made during the preparation of the raw material bearing, which de facto can never be eliminated 100%, there is a high probability of the formation of the C12A7 phase.

Remark 15: L129-L130: Please provide more detail for the following sentence: "The sintered samples were then cooled in air resulting in a cubic C3A phase (see Fig. 7 below)".

Response: Thank you for this comment. After removing the samples from the furnace, the resulting sinter was ground in an agate mill to a grain size corresponding to a 0.01 mm sieve mesh. The powder was then subjected to XRD analysis to confirm the presence of a cubic C3A phase.

Remark 16: L133-L134: Are they diluted or following a special method of preparation? Please provide details, such as the sample-to-curing medium ratio, the chemical composition of these media, etc.,

Response: Thank you for this comment. The biological materials in a quantity of 10 kg were diluted with 10 litres of distilled water. The aqueous solution thus prepared was subjected to maturation for a period of 14 days in a sealed plastic barrel. This information has been added to manuscript text.

Remark 17: L186: an FTIR spectrometer.

Response: The sentence has been corrected

Remark 18: L193: at a magnification of 350x.

Response: This has been corrected

Remark 18: L207: Exposure of tricalcium aluminate to a pig slurry, the preparation of slurry should be described.

Response: The biological materials in a quantity of 10 kg were diluted with 10 litres of distilled water. The aqueous solution thus prepared was subjected to maturation for a period of 14 days in a sealed plastic barrel. This information has been added to manuscript text in the section 2.3. Preparation of samples for testing

Thank you again for the opportunity to improve the quality of our manuscript. We hope that we have answered the Reviewer's remarks in a satisfactory manner.

                                                                                               Kind regards,

                                                                                               Authors
